

# Southern hemisphere bog persists as a strong carbon sink during droughts

Jordan P. Goodrich[1,2], Dave I. Campbell[1], Louis A. Schipper[1]

[1]School of Science, University of Waikato, Hamilton, Private Bag 3105, New Zealand
[2]Now at: Scripps Institution of Oceanography, UC San Diego, San Diego, CA 92093, USA

*Correspondence to*: Jordan P. Goodrich (jpgoodrich@ucsd.edu)

**Abstract.** Peatland ecosystems have been important global carbon sinks throughout the Holocene. Most of the research on peatland carbon budgets and effects of variable weather conditions has been done in Northern Hemisphere *Sphagnum*-dominated systems. Given their importance in other geographic and climatic regions, a better understanding of peatland
carbon dynamics is needed across the spectrum of global peatland types. In New Zealand, much of the historic peatland area has been drained for agriculture but little is known about rates of carbon exchange and storage in unaltered peatland remnants that are dominated by the jointed wire-rush, *Empodisma robustum*. We used eddy covariance to measure ecosystem-scale $CO_2$ and $CH_4$ fluxes and a water balance approach to estimate the sub-surface flux of dissolved organic carbon from the largest remaining raised peat bog in New Zealand, Kopuatai bog. The net ecosystem carbon balance
(NECB) was estimated over four years, which included two drought summers, a relatively wet summer, and a meteorologically average summer. In all measurement years, the bog was a substantial sink for carbon, ranging from 134.7 gC m$^{-2}$ yr$^{-1}$ to 216.9 gC m$^{-2}$ yr$^{-1}$, owing to the large annual net ecosystem production (-161.8 to -244.9 gCO$_2$-C m$^{-2}$ yr$^{-1}$). Annual methane fluxes were large relative to most Northern Hemisphere peatlands (14.2 to 21.9 gCH$_4$-C m$^{-2}$ yr$^{-1}$), although summer and autumn emissions were highly sensitive to dry conditions leading to very predictable seasonality according to
water table position. The annual flux of dissolved organic carbon was similar in magnitude to methane emissions but less variable, ranging from 11.7 to 12.8 gC m$^{-2}$ yr$^{-1}$. Dry conditions experienced during late summer droughts led to significant reductions in annual carbon storage, which resulted nearly equally from enhanced ecosystem respiration due to lowered water tables and increased temperatures, and from reduced gross primary production due to vapor pressure deficit-related stresses to the vegetation. However, the net C uptake of Kopuatai bog during drought years was large relative to even the
maximum reported NECB from Northern Hemisphere bogs. Furthermore, GWP fluxes indicated the bog was a strong sink for greenhouse gases in all years despite the relatively large annual methane emissions. Our results suggest that adaptations of *E. robustum* to dry conditions lead to a resilient peatland drought response of the NECB.

## 1 Introduction

Peatlands occupy a small fraction of the global land area (~2-3%) but store a large proportion of soil carbon (~50%)

(Gorham, 1991; Yu et al., 2010). Fluxes of carbon (C) in peatlands are sensitive to variations in weather and climate, but

large uncertainties are associated with scaling process knowledge to landscapes or regions (Baird et al., 2009), and with high

variability within and among sites (Bubier et al., 2003; Mastepanov et al., 2008; Treat et al., 2007). Furthermore, there is a

relative paucity of *in-situ* measurements for many globally important peatland types (Christensen, 2014). The majority of

global peatland area is located in boreal to Arctic regions of the Northern Hemisphere, which has led most of the research on

peatland C exchange to focus on these regions (Lafleur, 2009). However, peatlands also make up an important component of

the landscape in other regions where much less is known about the size of C stocks, rates and variability of C fluxes, and



sensitivity of those fluxes to environmental and climatic change (Frolking et al., 2011; Lafleur, 2009; Limpens et al., 2008). Tropical and Southern Hemisphere peatlands are particularly under-represented in the literature, despite contributing 10% of the global peatland area (Yu et al., 2010) and potentially 50% of the global $CH_4$ emissions in a given year (Bousquet et al., 2011). Furthermore, many of the peatlands in these regions are under pressure of changing land use and increased fire

frequency (e.g., Page and Hooijer, 2016; Perry et al., 2014) while minimal baseline data are available to assess the potential changes to regional C budgets. Given the large amount of C stored in peatland ecosystems, anthropogenic impacts on peatland function, and the potential for positive feedbacks between peatland C fluxes and changing climate, a better understanding of C exchange processes along the full spectrum of peatland types is needed.

In New Zealand and Australia, members of the exclusively Southern Hemisphere Restionaceae family of rush-like vascular

plants are the predominant peat-formers in low altitude mires (Agnew et al., 1993), and relatively little is known about their C exchange properties, either on an individual or ecosystem scale (Meney and Pate, 1999; Goodrich et al., 2015a). In the Waikato region of New Zealand, peatlands occupy about 5% of the land area, the majority of which has been drained for dairy pasture (McGlone, 2009; Pronger et al., 2014). The draining of peatlands leads to subsidence associated with enhanced mineralization and compaction (Pronger et al., 2014; Schipper and McLeod, 2002), both of which result in large C losses of

the affected area. However, intact peatland remnants adjacent to drained pastures seem to remain strong annual sinks for $CO_2$ despite artificially lowered water tables (Campbell et al., 2014). The vascular plant-dominated vegetation in New Zealand peatlands has very conservative evaporation rates and high canopy resistance to water vapour exchange during dry, sunny periods (Campbell and Williamson, 1997). This behaviour also constrains gross primary production (GPP) when vapour pressure deficit is high (Goodrich et al., 2015a). Lowered water tables during drought conditions lead to reduced methane

fluxes, which remain low for months after water table recovery, substantially reducing annual $CH_4$ emissions during drought years (Goodrich et al., 2015b). However, we do not yet know the full extent of the response of the net ecosystem C balance (NECB), – and its components – in unaltered New Zealand peatlands to dry versus wet conditions on seasonal to annual timescales. Therefore we have little basis for predicting NECB changes in these systems when environmental changes are imposed from neighbouring land use intensification (e.g., Fetzel et al., 2014) and with the potential for increasing severity of

summer droughts due to climate change (Perry et al., 2014; Dai, 2013; Trenberth et al., 2014).

Northern Hemisphere peatlands can shift from annual sinks to sources of $CO_2$ in response to drought (Arneth et al., 2002), and drought-induced lowering of the water table is generally the most important driver of inter-annual variability in peatland C exchange (Gažovič et al., 2013; Herbst et al., 2011; Olson et al., 2013). Drought response of peatlands is also complicated by the potential for different effects on bogs compared to fens. For example, the sensitivity of net C exchanges to water table

depth under relatively normal ranges has been shown to be higher in bogs than fens (Lindroth et al., 2007), but water table and drought effects in non-*Sphagnum*-dominated peatlands are not well studied (Fritz et al., 2011; Cooper et al., 2015; Goodrich et al., 2015b). Furthermore, the relative impact of drought on each NECB component is not uniform across peatland types and may vary by dominant vegetation, litter quality, peatland hydrology, growing season length, nutrient status, or timing of drought (Bubier et al., 2003; Lund et al., 2012; Sulman et al., 2010). Expanding the coverage of peatland



C flux observations during drought to globally distinct vegetation types may aid our ability to determine common features and responses that lead to increased C losses or those that minimize drought effects.

We estimated the net ecosystem C balance (NECB) at a raised ombrotrophic bog in New Zealand over four years that included one of the most extreme meteorological droughts in the past 70 years (Porteous and Mullan, 2013). We used continuous eddy covariance (EC) measurements of $CO_2$ and $CH_4$ flux and a water balance approach to estimate DOC export in order to calculate monthly and annual budgets of each C flux component and to determine the main drivers of variability among years. Carbon flux measurements and estimated NECB from this site extend the range of climatic zones represented in peatland literature as well as add information on the response of a distinctive plant functional type to a wide range of environmental conditions. We also aim to highlight some useful parallels and contrasts between this globally unique peatland system and the much better represented Northern Hemisphere peatlands as well as the growing body of tropical peatland literature, with respect to drought effects on C fluxes.

## 2 Methods

### 2.1 Site description

Kopuatai bog is located in the Hauraki Plains of the North Island/Te Ika a Māui, New Zealand/Aotearoa (37.387S, 175.459E). This ombrotrophic raised bog is the largest remaining unaltered peatland (~90 km$^2$) in the country since the majority of New Zealand wetlands have been drained, primarily for agriculture (McGlone, 2009). The vegetation at the site is dominated by the jointed wire rush, *Empodisma robustum* (Wagstaff and Clarkson, 2012), which forms a dense canopy (mean height ~0.8 m) of interwoven live and dead stem material. Total green plant area index (GAI) at the site is $1.32 \pm 0.32$ m$^2$ m$^{-2}$ and standing brown living plant material and dead litter amounts to $1.33 \pm 0.54$ kg m$^{-2}$ (Goodrich et al., 2015a), primarily contributed by *E. robustum*. Other vegetation cover includes *Gleichenia dicarpa*, sedges *Machaerina teretifolia* and *Schoenus brevifolius,* especially in wetter zones, and clusters of small shrubs 1-2 m in height, *Leptospermum scoparium* and *Epacris pauciflora*, sparsely scattered throughout the study area. *Sphagnum* and other mosses occur rarely throughout the peatland where coverage of the dominant vegetation is sparse and light penetrates to the surface, however the primary peat forming material is *E. robustum* roots (Agnew et al., 1993). These roots form negatively geotropic clusters covering the surface, and serve a similar nutrient capture and water holding capacity role as *Sphagnum* in Northern Hemisphere peatlands (Agnew et al., 1993; Clarkson et al., 2009a).

### 2.2 Eddy covariance $CO_2$, $H_2O$, and $CH_4$ flux measurements

We measured net ecosystem exchange of $CO_2$ ($F_{CO2}$) and $H_2O$ (latent heat flux, LE) using the eddy covariance (EC) technique from 19 November 2011 to 31 December 2015, while methane flux ($F_{CH4}$) measurements began 4 February 2012. Our EC instrumentation was mounted on a 4.5 m tall triangular lattice tower and included a sonic anemometer (CSAT3, Campbell Scientific Inc., Logan, Utah, USA), an open path $H_2O/CO_2$ analyzer (LI-7500, LI-COR Biosciences Inc., Lincoln,




NE, USA) and an open-path $CH_4$ analyzer (LI-7700, LI-COR Inc.). Sensors were mounted on a horizontal boom approximately 1.5 m from the face of the tower with uninterrupted fetch extending >500 m in all directions and relatively uniform canopy height and negligible slope over that distance. Data were collected at 10 Hz using a CR3000 datalogger (Campbell Scientific Inc.) and stored on a memory card.

Fluxes were processed with an averaging interval of 30 minutes using the EddyPro software (v5.1.1, LI-COR Inc.). Time lags between the wind and scalar concentration time series were removed by covariance maximization. A fully analytic approach was chosen for correction of low-pass (Moncrieff et al., 1997) and high-pass filtering (Moncrieff et al., 2005) and the standard Webb et al. (1980) method was applied to compensate for the effects of air density fluctuations on both $F_{CO2}$ and $F_{CH4}$. A double-axis rotation was applied for sonic tilt correction and the concentration time series were de-trended by

block averaging. Spikes in the high frequency data were removed according to Vickers and Mahrt (1997). During EddyPro flux processing and computation, we implemented the Göckede et al. (2006) method to assign quality control (QC) flags to each 30-min flux, based on tests for well-developed turbulence and steady state conditions (flags 1 – 5, with 1 being best quality). We corrected $F_{CO2}$ for storage changes in the layer below the EC instruments based on changes in 30-minute $CO_2$ concentrations measured by the LI-7500, but did not adopt this procedure for $F_{CH4}$ because it made very little difference to

daily and longer-term sums, instead introducing more noise. Ancillary measurements included incoming total and diffuse photosynthetic photon flux density (PPFD) (BF5 Sunshine Sensor, Delta-T Devices Ltd., Cambridge, United Kingdom) above the canopy (~1.2 m above peat surface); incoming and outgoing shortwave and long wave radiation fluxes and canopy surface temperature ($T_{surf}$) (NR01, Hukseflux, Delft, The Netherlands) at 2 m height on a secondary mast 5 m northwards of the EC tower; air temperature ($T_{air}$) and vapour pressure ($e_{air}$) (fully aspirated HMP 155, Vaisala, Helsinki, Finland) at 4.25 m

above the surface. Water table depth (WTD) was measured using a submersible pressure sensor (WL1000W, Hydrological Services, NSW, Australia) suspended within a 1.5 m-long dipwell constructed from 50 mm diameter PVC slotted along its length, anchored to a wooden board laced to the peat surface. Rainfall was measured with a tipping bucket rain gauge (TB3, Hydrological Services, NSW Australia).

**2.3 Quality control, gap filling, and flux partitioning**

Fluxes assigned QC flag values > 1 were discarded from the analysis. Data were then filtered for insufficient atmospheric turbulence using a threshold for friction velocity ($u_* < 0.15$ m s$^{-1}$), below which all flux data were discarded. We chose this cut-off after calculating annual sums of $F_{CO2}$ and $F_{CH4}$ using a range of $u_*$ thresholds and determining the value at which the annual sums stabilized, following Loescher et al. (2006). In addition, fluxes were discarded when the associated wind directions fell within a 55° sector that included the tower and site infrastructure.

Gaps in all fluxes were filled using artificial neural networks (ANN). The ANN used to fill gaps in $F_{CH4}$ was described in Goodrich et al. (2015b). Given that the $F_{CH4}$ measurements began 4 February 2012, we used the ANN to estimate January 2012 fluxes and to obtain a full four-year dataset. Gaps in the $F_{CO2}$ time series were also filled using an ANN approach (Papale and Valentini, 2003), separately for daytime (PPFD > 5 µmol m$^{-2}$ s$^{-1}$) and nighttime (PPFD ≤ 5 µmol m$^{-2}$ s$^{-1}$). The



nighttime ANN consisted of nine input nodes including air temperature ($T_{air}$), peat temperature at 50 mm below the surface ($T_{peat}$), water table depth, four fuzzy datasets representing season, one fuzzy dataset representing the year of study period, and an offset node. The daytime ANN had inputs of PPFD, $T_{air}$, canopy surface temperature ($T_{surf}$), atmospheric vapour pressure deficit (VPD), VPD within the upper canopy estimated using measured $T_{surf}$ and $e_{air}$ (VPD$_{surf}$), WTD, and the same

fuzzy datasets described for the nighttime ANN. Both nighttime and daytime ANNs included four hidden nodes, and sigmoid transfer functions were applied to the weighted sums from the hidden and output layers (Dengel et al., 2013; Papale and Valentini, 2003). Since each neural network run gives a unique result, both day time and nighttime ANN models were trained and fitted 25 times and the median values were used to fill gaps (Knox et al., 2014). Similarly, the ANN used to fill gaps in daytime LE consisted of six input variables (horizontal wind speed, $T_{air}$, VPD, $T_{surf}$, net radiation ($R_n$), and VPD$_{surf}$),

and eight fuzzy variables describing season of year and time of day. Nighttime gaps in LE were filled with ANN output driven by $T_{air}$, VPD, VPD$_{surf}$, $R_n$, and horizontal wind speed, and three fuzzy variables describing season of year.

To partition $F_{CO2}$ into gross primary production (GPP) and ecosystem respiration (ER), we estimated daytime ER by applying the nighttime ANN to daytime driver data (Desai et al., 2008; Oikawa et al., 2017). Oikawa et al. (2017) showed that results from flux partitioning based on neural networks behaved similarly to those based on the Reichstein et al. (2005)

approach in an alfalfa field. However, since nighttime respiration is often greater than during daytime (Kok effect) (Oikawa et al., 2017, Kok, 1949), both approaches may over-estimate both partitioned GPP and ER. Nonetheless, for this study, nighttime GPP was assumed to be zero and daytime GPP was estimated by subtracting modelled daytime ER from gap-filled $F_{CO2}$. We use the term net ecosystem production (NEP) to refer to monthly and annual summed $F_{CO2}$, representing the difference between GPP and ER, so that NECB = NEP - $F_{CH4}$ - $F_{DOC}$, and positive NECB indicates C uptake by the

ecosystem.

**2.4 DOC export**

A detailed description of the methods used for estimating C loss via dissolved organic C (DOC) export in subsurface water ($F_{DOC}$) was given by Sturgeon (2013) and is the subject of Campbell et al., (in preparation). Briefly, during 2012, DOC was sampled monthly at nine sites across the EC footprint, at three peat depths, by extracting water from PVC wells sampling

depth ranges 0 – 0.3 m, 0.3 – 0.6 m, and 0.6 – 1.0 m. The concentration of DOC in water samples was determined with a TOC-VCSH analyser (Shimadzu, Kyoto, Japan). Monthly water seepage from the EC footprint was estimated with a water balance approach: $Q = P - E - \Delta S$, where $P$ is rainfall, $E$ is evaporation and $\Delta S$ is change in water storage (all with units mm). Daily totals of $E$ were calculated from gap-filled time series of 30-min LE, and $\Delta S$ was calculated at monthly time-steps from changes in water table depth multiplied by peat specific yield.  $F_{DOC}$ for 2012 was initially calculated as the

product of depth-weighted mean monthly DOC concentration and monthly $Q$. There was a strong relationship between monthly ($P – E$) and $F_{DOC}$ (Sturgeon, 2013; Campbell et al., in preparation), so monthly $F_{DOC}$ for the whole study period was calculated from this relationship (Fig. S1).



### 2.5 Uncertainty estimates

Random uncertainty for each half-hourly value of $F_{CH4}$ and $F_{CO2}$ was estimated based on whether the value was measured or gap-filled (Dragoni et al., 2007). For measured values we applied the 'paired-days' approach of Hollinger and Richardson (2005) for which the difference between matching half-hourly fluxes (either $\Delta F_{CO2}$ or $\Delta F_{CH4}$) on adjacent days were

examined if environmental data were similar (PPFD within 75 µmol m$^{-2}$ s$^{-1}$, $T_{air}$ within 3 °C, wind speed within 1 m s$^{-1}$). To apply this approach to $F_{CH4}$, additional constraints were added for WTD (within 5 mm) and $T_{peat}$ (within 2 °C) given their influence on CH$_4$ production and flux (Goodrich et al., 2015b). Double exponential distributions (maximum likelihood = $\frac{1}{2}\beta e^{-|x-\mu|/\beta}$, where β is the mean of absolute deviations of the samples, and µ is the sample mean) were fitted to $\Delta F_{CO2}$ and $\Delta F_{CH4}$ binned by flux magnitude and the uncertainty of each measured half-hourly flux value ($\sigma_m = (\sqrt{2})\beta$) was

determined as a function of the mean flux between the measurement pairs (Dragoni et al., 2007; Hollinger and Richardson, 2005). We utilized the residuals from the 25 ANN simulations for $F_{CH4}$ and $F_{CO2}$ to estimate uncertainty owing to the gap-filling approach. These residuals were normally distributed so the standard deviations ($\sigma_{gf}$) were determined as functions of the gap-filled flux magnitudes (Dragoni et al., 2007). Uncertainty in monthly $F_{DOC}$ was calculated as the 95% confidence intervals around the predicted value based on monthly P-E (Fig. S1). Monthly uncertainty values were then combined in

quadrature to obtain annual uncertainty estimates.

### 3 Results

### 3.1 Meteorological and hydrological conditions

Mean annual air temperatures were 13.3, 14.0, 13.9, and 13.6 °C for 2012– 2015, respectively, compared to the 30-year mean of 13.7 °C at an official climate station 11 km to the east of the study site (New Zealand National Institute for Water

and Atmospheric Research, Taihoro Nukurangi). Annual totals of precipitation (100% rain) were 1153, 1105, 1086, and 1167 mm compared to the 30-year mean of 1232 mm. Despite annual rainfall being within 4% of the mean for all four years, the summer (Jan – Mar) rainfall sums in 2013 and 2014 were particularly low with 65 and 103 mm, respectively, compared to the much wetter 2012 summer (289 mm) and somewhat less extreme 2015 summer (176 mm) (Fig. 1). These precipitation patterns also manifest in late summer minima in water table depth, with 2013 exhibiting the lowest WTD of the

measurement period (~ 300 mm below the surface) (Fig. 1). Water table depths recharged to within ~50 mm of the peat surface each winter responding sharply to rainfall events.

### 3.2 Variations in NECB components

Annual NECB totals at Kopuatai bog were 210.2, 134.7, 143.3, and 216.9 gC m$^{-2}$ yr$^{-1}$ in the years 2012–2015, respectively (Table 1). GPP and ER were the largest terms in the budget for all years. Annual GPP totals were similar for 2012–2014

(ranging 791.3 to 815.3 gC m$^{-2}$ yr$^{-1}$), but larger in 2015 (880.5 gC m$^{-2}$ yr$^{-1}$) (Table 1). Monthly GPP was >20 gC m$^{-2}$ for

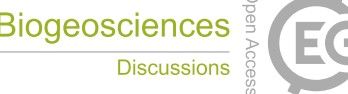



every month of the study period (Fig. 2a), indicating year-round growing conditions. Ecosystem respiration was roughly 10% lower in 2012 (570.5 gC m$^{-2}$ yr$^{-1}$) than in all other years (ranging 629.2 to 636.5 gC m$^{-2}$ yr$^{-1}$) (Table 1), primarily as a result of reduced respiration during the wet summer in 2012 with generally higher water table (Figs. 1d and 2b). The resulting annual totals of NEP were 244.9, 161.8, 169.9, and 243.7 gC m$^{-2}$ yr$^{-1}$ in 2012-2015 (Table 1).

Annual $F_{CH4}$ was a much smaller component of NECB than NEP, with emissions representing 21.9 (10%), 14.7 (11%), 14.9 (10%), and 14.2 (7%) gC m$^{-2}$ yr$^{-1}$ in 2012-2015 (Table 1) with much lower monthly fluxes during the drought months and subsequent slow recovery after water table recharge (Fig. 3a). Annual $F_{DOC}$ contributed a similarly small proportion but consistent fluxes of 12.8 (6%), 12.4 (9%), 11.7 (8%), and 12.6 (6%) gC m$^{-2}$ yr$^{-1}$ in 2012-2015 (Table 1). $F_{DOC}$ was the most variable flux from month to month (Fig. 3b) being driven primarily by the water balance (Fig. S1).

**3.3 Seasonal variation in $F_{CO2}$**

Seasonal variation in diel ensemble $CO_2$ fluxes was relatively constrained (Fig. 4). Despite significant differences in mid-day (hours 10 – 14) $CO_2$ uptake among seasons (ANOVA: F = 579.4, p < 0.001), the winter mean (-3.2 μmol m$^{-2}$ s$^{-1}$) was just 34% lower than summer mean uptake (-4.8 μmol m$^{-2}$ s$^{-1}$) (Fig. 4a, c). Mean nighttime (hours 20 – 5) $CO_2$ fluxes were also significantly different among seasons (ANOVA: F = 429.7, p < 0.001), with mean winter nighttime losses 40% lower than in

summer (Fig. 4a, c). The most substantial inter-annual deviations from mean $F_{CO2}$ patterns occurred in 2012 when summer mid-day $CO_2$ uptake was 34% greater (-5.9 μmol m$^{-2}$ s$^{-1}$) than the mean of the three other years (-3.9 μmol m$^{-2}$ s$^{-1}$) (Fig. 4a). Differences among years were also prominent in spring and autumn mid-day $CO_2$ uptake (Fig. 4b, d), where 2015 exhibited the largest uptake in both cases.

The bog switched from $CO_2$-C sink to $CO_2$-C neutral or source one month earlier in 2013 than any other year owing to the

drier conditions and elevated ER during the 2013 drought coupled with slightly lower GPP (Fig. 2b). However, the bog was a slight $CO_2$-C source for only two months during that year (2013) and neutral for a third month. During 2015, which was neither abnormally wet nor dry, NEP was positive for eleven months and neutral for one (Fig. 2c).

**3.5 Controls on ecosystem C fluxes**

Variation in monthly NECB was best described by a simple linear regression with monthly total PPFD, whereby the

ecosystem was a significantly stronger C sink during summer months than during winter months (Figs. 5, 6). As GPP was the largest gross term in the budget, the seasonal progression of NECB (Fig. 5) was generally similar to that of GPP and NEP (Fig. 2a,c), effectively resulting in light limitation of overall NECB at monthly timescales (Fig. 6). However, inter-annual differences in monthly NECB were driven by changes to both ER and GPP (Fig. 7).

To assess the drivers of ER and GPP, we isolated summer (December – February) and autumn (March - May) months since

differences in mean fluxes between dry and wet years were largest during these seasons (Figs. 2 and 4). Mean monthly ER was strongly driven by WTD and $T_{peat}$ (Table 2). Higher $T_{peat}$ led to higher respiration (Fig. 7a), while this enhancement in ER was also exacerbated by lowered WTD (vertical stratification of colors in Fig. 7a). Accounting for changes in both





variables improved the regression model, explaining about 86% of the variance in ER, over the simple models including only WTD or $T_{peat}$ (Table 2). The corresponding variation in mean monthly GPP among years was largely driven by total PPFD and VPD, whereby higher PPFD/VPD led to reduced GPP (Figure 7b). Since changes in VPD were closely correlated with changes in PPFD (Goodrich et al., 2015a), there were more subtle differences in regression results using one or the other or

both variables in explaining variance in GPP (Table 2) compared to the equivalent for ER. In addition, lowered water table *per se*, did not seem to impact GPP significantly (not shown). Although drier, warmer conditions had a larger (up to 20% increase) proportional impact on summer ER (increasing with lowered water tables and higher $T_{peat}$) than the 5-18% decrease in GPP (decreasing with higher VPD), the contribution of lowered GPP to the overall NECB during those months was similar to ER (Fig. 7) because of the larger relative magnitude of GPP (Fig. 2).

**4 Discussion**

**4.1 Peatland net ecosystem C balance**

Kopuatai bog was a strong C sink during four years with contrasting environmental conditions that included late summer droughts in 2013 and 2014. For all four measurement years, Kopuatai NECB was much larger (135 – 217 g C m$^{-2}$ yr$^{-1}$) than published Northern Hemisphere bog NECB estimates, which range from losses of 14 g C m$^{-2}$ yr$^{-1}$ to gains of 101 g C m$^{-2}$ yr$^{-1}$

(Dinsmore et al., 2010; Gažovič et al., 2013; Koehler et al., 2011; Nilsson et al., 2008; Olefeldt et al., 2012; Roulet et al., 2007).

The relative contributions of non-$CO_2$-C components to Kopuatai's NECB (~10% each) were comparable to those estimated in other peatland NECB studies (Koehler et al., 2011; Nilsson et al., 2008; Roulet et al., 2007). However, the relatively short season of C loss at Kopuatai was largely due to the mild climate that resulted in year-round growing conditions and

relatively large annual NEP (Table 1). This result is in agreement with Campbell et al. (2014), who found large annual NEP for a drainage-impacted New Zealand bog despite having similar peak summertime $CO_2$ uptake to Northern Hemisphere peatlands. The Campbell et al. (2014) study was conducted at Moanatuatua, a remnant bog with prevalence of the taller, late successional restiad species, *Sporadanthus ferrugineus* (giant cane-rush, Clarkson et al., 2004) in addition to *E. robustum*, resulting in greater mid-day and annual GPP than we observed at Kopuatai. However, mean nighttime $F_{CO2}$ (Fig. 4b) and

total ER during summer drought months at Kopuatai (Fig. 2b) were similar to those observed during summer at Moanatuatua bog, despite water tables reaching 80 cm below the surface there (Campbell et al., 2014) compared to < 30 cm below the surface at Kopuatai (Fig. 1). Similarly, Lafleur et al. (2005) showed that ER at the relatively dry Mer Bleue bog, in Ontario, Canada, was only weakly correlated to water table depth. The lack of increase in ER at Mer Bleue with dropping water tables may have been related to compensating factors of decreased respiration from desiccated surface *Sphagnum* offsetting

increased respiration of deeper heterotrophic microbial communities (Dimitrov et al., 2010). Our results suggest that lowered water tables increase ER at Kopuatai but there may be a limit to this increase, were water table to continue dropping. Further



enhancements in ER may require a shift away from *E. robustum*, with its conservative evaporation regime (Campbell and Williamson, 1997), to vegetation with higher water use.

Growing seasons at Northern Hemisphere peatlands are generally bounded by frozen or snow-covered winters but year-round GPP > 0 has been reported at an Atlantic blanket bog, Glencar, subject to a relatively mild, maritime climate (McVeigh et al., 2014; Sottocornola and Kiely, 2010). However, mean summertime peak GPP and ER at Kopuatai (114.4 gC $m^{-2}$ $mo^{-1}$ and 73.0 gC $m^{-2}$ $mo^{-1}$, respectively, Fig. 2) were substantially higher than reported for Glencar (63.7 gC $m^{-2}$ $mo^{-1}$ and 38 gC $m^{-2}$ $mo^{-1}$, respectively) (McVeigh et al., 2014), which may be partly due to the lower peak LAI there (~0.6 $m^2$ $m^{-2}$) compared to Kopuatai (1.3 $m^2$ $m^{-2}$) (Goodrich et al., 2015a), as well as less solar radiation at the higher latitude Irish site. In contrast, a moderately rich treed fen in Western Canada with higher LAI (2.61 $m^2$ $m^{-2}$) had larger peak GPP and ER than we found at Kopuatai, leading to similar annual totals (713 gC $m^{-2}$ $yr^{-1}$ and 596 g C $m^{-2}$ $yr^{-1}$, respectively) despite a shorter, 6 month growing season (Syed et al., 2006). Lund et al. (2010) showed that LAI and growing season length explained a large proportion of the variance in NEP and its components across a range of northern peatlands. Our results from Kopuatai bog are consistent with the relationship between summertime NEP and LAI established by Lund et al. (2010), given the relatively high LAI and NEP measured here.

## 4.2 Drought effects on Kopuatai NECB and global context

In years with summer/autumn drought (2013 and 2014), including one of the most severe and widespread meteorological droughts in New Zealand in the past 70 years (Porteous and Mullan, 2013), NECB at Kopuatai bog was reduced by roughly 30-40% compared to the relatively wet or meteorologically 'normal' years (2012 and 2015). However, the bog was still a strong C sink during early drought months in both 2013 and 2014 (Fig. 5) and overall during the drought years (Table 1). Total GPP in January 2012 and December 2015 were higher than for any other months during the study period (Fig. 2), which was likely caused by the low, but still saturating, PPFD and the associated low VPD conditions (Goodrich et al., 2015a). This also fit within a general pattern whereby the largest monthly GPP values occurred during saturating light levels but reduced VPD (Fig. 7b). Similarly, Aurela et al. (2007) showed that GPP at a sedge fen in Finland was relatively unchanged during a drought summer, although rates of uptake during clear-sky afternoons within drought months were suppressed due to high VPD, contributing a small percentage of the overall drought-induced reduction in NEP.

Some Northern Hemisphere peatlands shift from annual (or growing season) sinks to sources of $CO_2$ in response to dry conditions (Alm et al., 1999; Joiner et al., 1999; Shurpali et al., 1995). Reduction in peatland NEP during dry conditions can result from reduced GPP, increased ER, or a combination of both. $F_{CH4}$ tends to be reduced during dry years (Brown et al., 2014; Moore et al., 2011), while observations of $F_{DOC}$ during dry years in different peatland types are less conclusive (Koehler et al., 2011; Roulet et al., 2007). Our results suggest that both $F_{CH4}$ and $F_{DOC}$ were lowered during dry months but that these changes contributed only slightly to the overall NECB response to dry conditions.

The relative response of GPP and ER to dry conditions has important implications for the future C sink status of peatlands under changing climates (Wu and Roulet, 2014). However there is no consensus on whether changes in GPP or ER are more





important to the peatland NEP drought response (Lafleur, 2009) and very little data have yet been obtained in tropical and Southern Hemisphere systems.

Most studies reporting peatland $CO_2$ fluxes during relatively dry conditions attribute some portion of the NEP reduction to an increase in ER, including in the tropics (Hirano et al., 2009) and therefore, the ER response at Kopuatai was expected.

However, the reported effects of a lowered water table *per se* on GPP are more varied in the literature. Differences in GPP among years due to WTD changes at our site were small, indicating relative insensitivity in photosynthetic uptake of *E. robustum*-dominated peatlands to summer water table drawdown even during droughts. Some researchers have reported relatively low sensitivity of annual GPP to lowered water tables, often due to compensating factors that allowed NEP to recover despite either temporary reductions in GPP or increases in ER (Aurela et al., 2007). The only peatland in which GPP

has been reported to increase in response to drought conditions was a treed, moderately rich fen in Western Canada (Cai et al., 2010; Flanagan and Syed, 2011). However, that site may have been within a successional phase toward increased tree growth and more above ground C allocation (Flanagan and Syed, 2011). The late-successional New Zealand peatland species, *Sporadanthus ferrugineous* (giant cane-rush), has deeper roots (Clarkson et al., 2009b) and higher above-ground biomass than the mid-successional *E. robustum* (Thompson et al., 1999), dominant at our study site. The large annual $CO_2$

sink strength reported from the much drier Moanatuatua bog, dominated by *S. ferrugineus* (Campbell et. al., 2014), highlights the need for future work in New Zealand peatlands to investigate the potential shift in C allocation from below ground (accumulating root biomass and peat) to above ground (stem and shoot biomass) resulting from succession or disturbance, such as long-term lowering of the water table.

Peatlands in which *Sphagnum* mosses contribute significantly to ecosystem GPP are particularly sensitive to dry conditions

(Shurpali et al., 1995; Alm et al., 1999; Arneth et al., 2002; Bubier et al., 2003; Lafleur et al., 2003), which is likely due to the inability of *Sphagnum* to control capitulum moisture content when water tables drop (Laitinen et al., 2008). Sulman et al. (2010) observed opposite responses of fens and bogs to inter-annual differences in water tables, conjecturing on the importance of relative *Sphagnum* cover in accounting for the observed differences. However, sites dominated by vascular vegetation can also exhibit reduced GPP with lowered water tables, and the magnitude of the response may depend on timing

of dry conditions. Joiner et al. (1999) found that a late summer drought led to early autumn senescence of the vascular vegetation at a fen site in Manitoba, Canada, while ER remained steady until temperatures dropped. In contrast, Griffis et al. (2000) showed that dry periods during the early growing season in a sub-arctic fen, while plants were developing, led to substantially reduced GPP relative to wetter years and impacted the whole growing season $CO_2$ uptake, such that the ecosystem was a source of $CO_2$. Lund et al. (2012) showed a very similar effect of a springtime-initiated drought on GPP at a

Swedish raised bog, where plant development and moss biomass accumulation were suppressed, impacting NEP over the course of the year and causing the ecosystem to act as an annual source of $CO_2$. In contrast, at the same site, a mid-summer drought did not have the same effect on the vegetation and only ER was affected (increased) (Lund et al., 2012).

Dominant vegetation type is clearly a critical factor in determining peatland NEP and NECB response to dry conditions. Timing of drought is also important but the impact of timing seems dependent on vegetation type and few examples of early




spring droughts are available in published peatland NEP records. However, Kupier et al. (2014) used mesocosms from a raised bog to demonstrate that peatland plant functional types determined when the peatland shifted from $CO_2$ sink to $CO_2$ source in response to drying.

Vascular plants are better adapted to functioning during dry conditions, given their ability to control water loss through stomata (Körner, 1995) and to access water with deeper roots. Furthermore, evergreen species (e.g., ericaceous shrubs and restiads) seem to be particularly resilient to drought stress. *E. robustum* peatlands may be especially well-equipped for drought given the mulch-like layer of dead stem material that accumulates above the surface, partially contributing to reduced evaporation rates during dry conditions (Campbell and Williamson, 1997).

**4.3 Radiative forcing of Kopuatai bog greenhouse gas budget**

The relative importance of peatland $CH_4$ emission or $CO_2$ uptake to net climate forcing by an ecosystem ultimately depends on the timescale of interest and the relative flux magnitudes. This is often assessed using the global warming potential (GWP) approach (IPCC, 1990). If the standard 100-year GWP factor for $CH_4$ (28, IPCC, 2013) is applied to the annual C gas fluxes from Kopuatai bog, the result suggests that this peatland had a net cooling effect on the atmosphere during the four measurement years. GWP fluxes were -78.5, -43.2, -65.5, and -355.7 $gCO_2$-equivalents $m^{-2}$ $yr^{-1}$, in 2012-2015, respectively (negative sign convention here indicates a net greenhouse gas sink). This is not a surprising conclusion considering the large annual NEP even during drought years compared to Northern Hemisphere bogs (Roulet et al., 2007) and the very deep peat deposits that have accumulated over Kopuatai bog's 14,000-year development (Newnham et al., 1995), both of which are associated with cooling the atmosphere through consistent yearly net removal of atmospheric C. Others have estimated that annual peatland $CH_4$ fluxes can sometimes offset the C gains from net $CO_2$ uptake, depending on latitude and environmental conditions in a given year (Roulet, 2000; Crill et al., 2000; Whiting and Chanton, 2001; Friborg et al., 2003). However, Frolking et al., (2006) showed that from the time of peatland formation, the sustained $CH_4$ emissions dominate the radiative forcing signal for only about 50-100 years before the $CH_4$ effect stabilizes due to a relatively short atmospheric lifetime (~12 years), while the $CO_2$ uptake-effect continues to accumulate, leading to net atmospheric cooling.

Although abrupt changes in peatland radiative forcing may be possible through high $CH_4$ emissions because of its high GWP, changes in $CO_2$ dynamics, while dampened in the short-term, are much longer-lasting (Frolking and Roulet, 2007). Losses of DOC may, however, add to the climate warming potential of a peatland depending on the fate of this C. The contribution of Kopuatai's DOC losses to the GWP would depend on the eventual loss as either downstream $CO_2$ or $CH_4$, but is likely small compared to the measured $CH_4$ fluxes. In some peatland catchments, DOC lost from peatland margins is bubbled to the atmosphere as $CO_2$ during transport in neighboring streams adding substantially to the overall C loss (Billet et al., 2015), which would contribute to a warming effect.

Radiative forcing considerations have important implications for peatland restoration efforts because the C balance, and thus the ratio of $CH_4$ emission to $CO_2$ uptake, should be considered a key aspect of a functioning peatland. For example, loss of C due to peatland drainage for cultivation in New Zealand (e.g., Pronger et al., 2014) likely has a profound impact on the



relationship between those peatlands and the climate system (Frolking et al., 2014; Frolking and Roulet, 2007; Frolking et al., 2006). Furthermore, Campbell et al. (2015) showed that grazing on drained peatlands can result in 190 gC m$^{-2}$ yr$^{-1}$ as $CO_2$ lost to the atmosphere and potentially near 300 gC m$^{-2}$ yr$^{-1}$ if the full NECB is accounted for. Thus the transient warming impact of $CH_4$ emissions upon re-wetting/re-establishing a peatland during restoration is trivial over the long run compared

to the need to restore the peatland's ability to accumulate C. Furthermore, Shoemaker and Shrag (2013) illustrated the dangers of over-valuing the climate impact of $CH_4$ compared to $CO_2$ if the ultimate goal is to slow the warming effects of anthropogenic activities.

## 5 Conclusions

We have shown that a warm temperate bog in New Zealand dominated by the vascular plant, *E. robustum*, was a strong C

sink even during drought years. Our results from Kopuatai bog extend the coverage of ecosystem-scale C response to a globally unique peatland plant functional type and provide insight into the role of plants in the drought response of peatlands in general. Although peak GPP was reduced during dry summer days and ER was enhanced during drought months, the overall effect was not large enough to shift the ecosystem to being a $CO_2$ source over the course of a dry summer/autumn. Furthermore, the importance of summer NEP to annual totals was reduced due to the year-round growing conditions. The

drought resilience of Kopuatai bog in terms of reduced, but still relatively large, annual carbon uptake, also provides insight into the existence of these peatlands in a climatic setting that would not generally be considered conducive to peatland development and persistence given the often negative summer water balance and warm annual temperatures (McGlone, 2009). The negative feedback between the dry conditions and lower evaporation rates (Campbell and Williamson, 1997), while reducing GPP, helps maintain high water tables, which may limit respiration losses of C and maintain plant

functioning.
Non-$CO_2$-C losses did not contribute to the drought-induced decreases in C sink strength of Kopuatai bog, as both $F_{CH4}$ and $F_{DOC}$ were lowest during dry months. While $F_{CH4}$ at Kopuatai is large relative to Northern Hemisphere peatlands and should be considered an important component of the greenhouse gas balance of the bog, the ecosystem persisted as a net greenhouse gas sink, according to the GWP approach, during both relatively wet and dry years covered in this study.

**Acknowledgements**

We thank the New Zealand Department of Conservation for granting us a permit to carry out research within Kopuatai Reserve, and Murray, Angela and Phil Brewster for access across their farmland. Dean Sandwell, Aaron Wall, Chris Morcom, and Chris Eager are thanked for their technical support. We also acknowledge Te Kupenga O Ngati Hako for supporting the presence of our research site at Kopuatai. Funding for this research included Landcare Research under



Ministry for Science and Innovation contract C09X1002 and the University of Waikato for equipment and scholarship funding.

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

10        Maximum. Geophysical Research Letters, 37(13).




**Table 1. Annual carbon balance and component fluxes (± estimated uncertainties, see Methods) at Kopuatai bog from 2012 to 2015 (all units are gC m$^{-2}$ yr$^{-1}$).**

| Year | CO$_2$-C | | | Non-CO$_2$-C | | |
|---|---|---|---|---|---|---|
| | GPP | ER | NEP | $F$CH$_4$ | $F$DOC | NECB |
| 2012 | 815.3 (±8.6) | 570.5 (±9.7) | 244.9 (±7.2) | 21.9 (±0.4) | 12.8 (±0.7) | 210.2 (±14.9) |
| 2013* | 791.3 (±8.1) | 629.2 (±13.3) | 161.8 (±12.4) | 14.7 (±0.4) | 12.4 (±0.7) | 134.7 (±19.9) |
| 2014 | 799.4 (±9.6) | 629.2 (±14.2) | 169.9 (±14.1) | 14.9 (±0.4) | 11.7 (±0.7) | 143.3 (±22.2) |
| 2015 | 880.5 (±10.1) | 636.5 (±19.0) | 243.7 (±14.1) | 14.2 (±0.3) | 12.6 (±0.7) | 216.9 (±25.7) |

*Extreme drought year

5   **Table 2. Regression statistics for comparison of multiple and single-driver linear models explaining summertime and autumn monthly ER and GPP over the four measurement years. Root mean square error (RMSE) and Akaike's informatin criterion (AIC) are given as measures of model error and relative quality.**

| Model | R$^2$ | RMSE | AIC |
|---|---|---|---|
| ER ~ WTD | 0.20 | 50.6 | 166.2 |
| ER ~ $T_{peat}$ | 0.67 | 21.2 | 145.3 |
| ER ~ WTD + $T_{peat}$ | 0.86 | 9.3 | 126.5 |
| GPP ~ PPFD | 0.80 | 78.0 | 115.3 |
| GPP ~ VPD | 0.62 | 147.1 | 115.2 |
| GPP ~ PPFD + VPD | 0.83 | 69.9 | 116.5 |




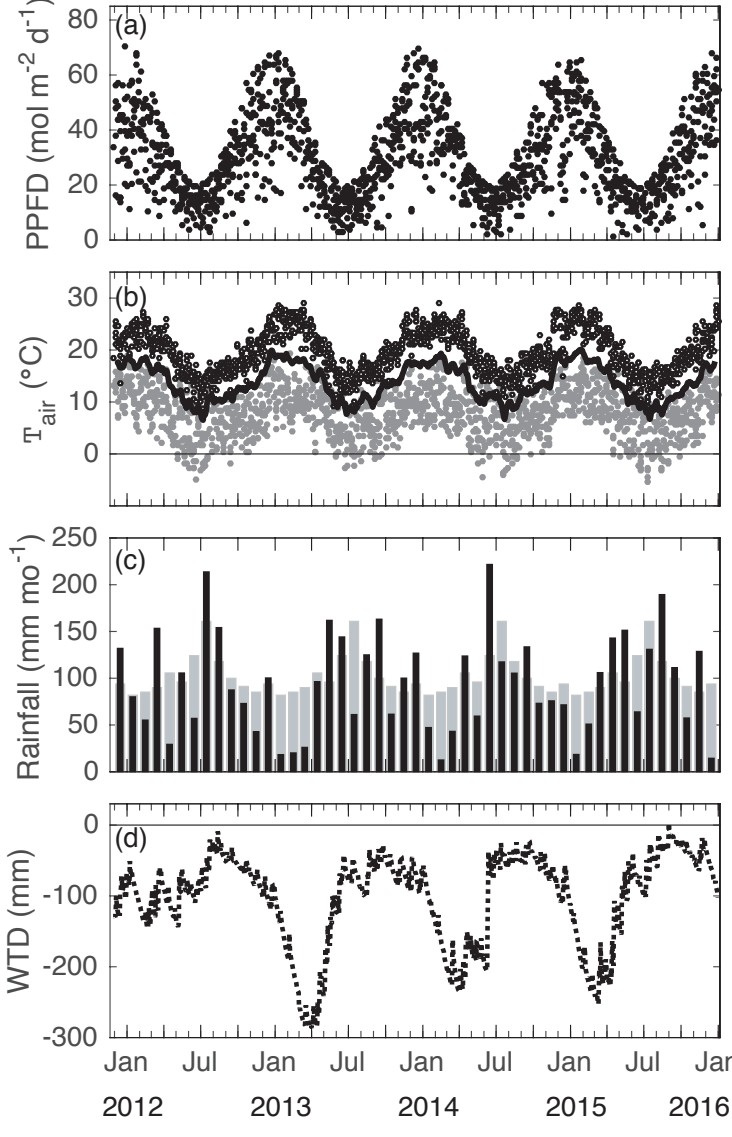

**Figure 1: Meteorological and hydrological variables at Kopuatai bog over the four measurement years. (a) Daily total incoming photosynthetic photon flux density (PPFD), (b) daily minimum (gray dots), maximum (black dots), and 15-day running mean air temperature ($T_{air}$) (line), (c) monthly total rainfall (black bars) and monthly climatologies (1980-2010) taken from a nearby climate station (gray bars), (d) daily mean water table depth (zero line is the peat surface).**



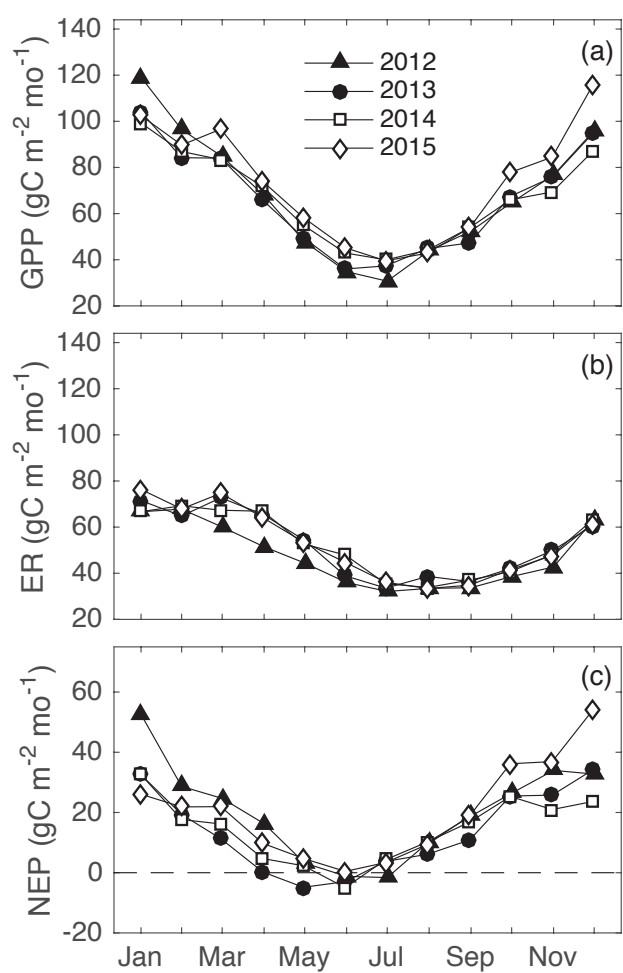

**Figure 2. Monthly CO$_2$-C flux components at Kopuatai bog over the four measurement years. (a) Gross primary production (GPP), (b) ecosystem respiration (ER), (c) net ecosystem production (NEP). Note the different y-axis scale in (c).**



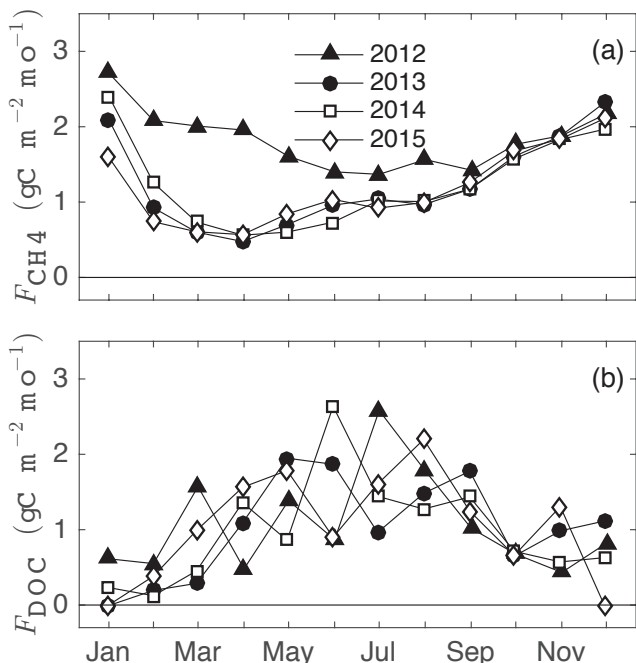

**Figure 3. Monthly total non-CO₂-C flux components at Kopuatai bog over the four measurement years. (a) Methane flux ($F_{CH4}$), (b) dissolved organic carbon export ($F_{DOC}$).**





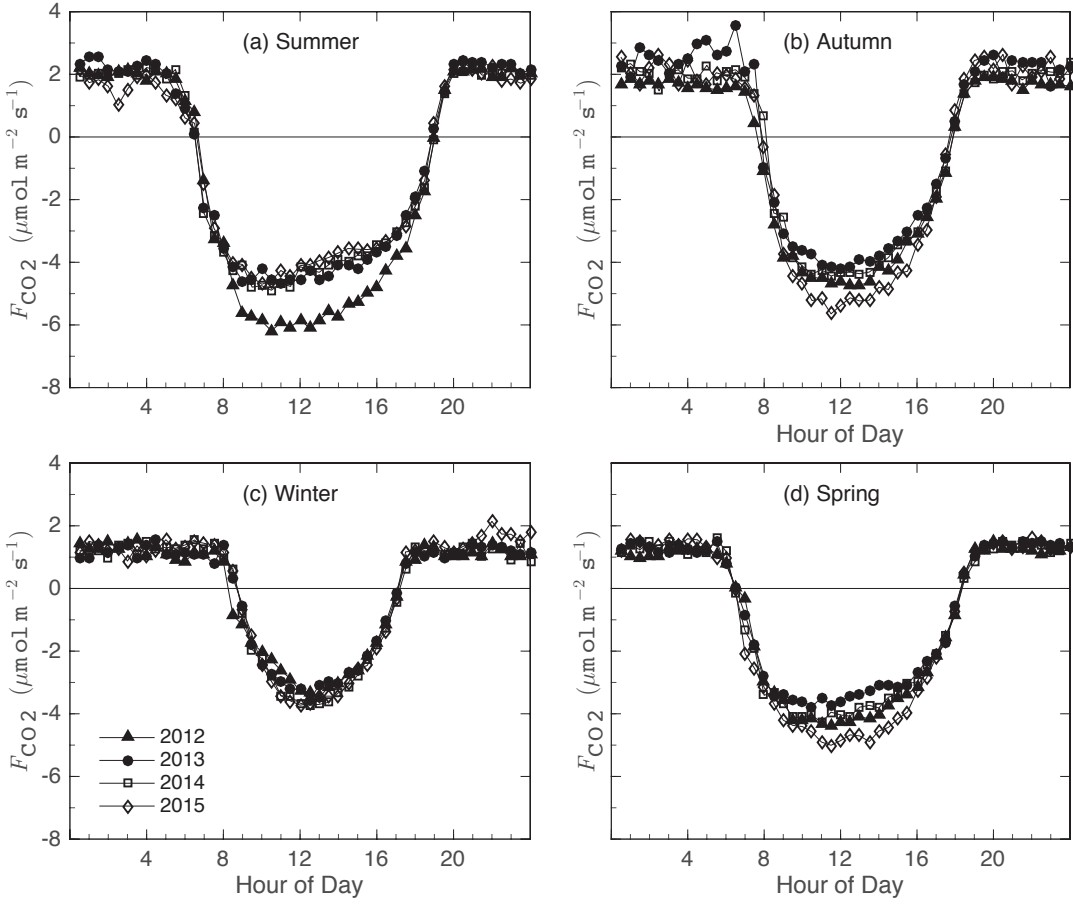

**Figure 4.** Seasonal ensemble average $F_{CO_2}$ (measured data) for (a) summer, (b) autumn, (c) winter, and (d) spring over the four measurement years.



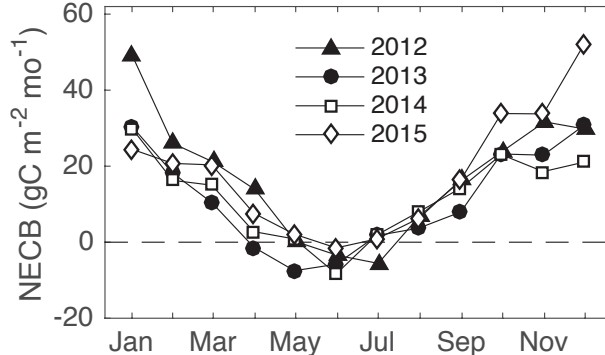

**Figure 5.** Monthly total net ecosystem carbon balance (NECB) at Kopuatai bog over the four measurement years.

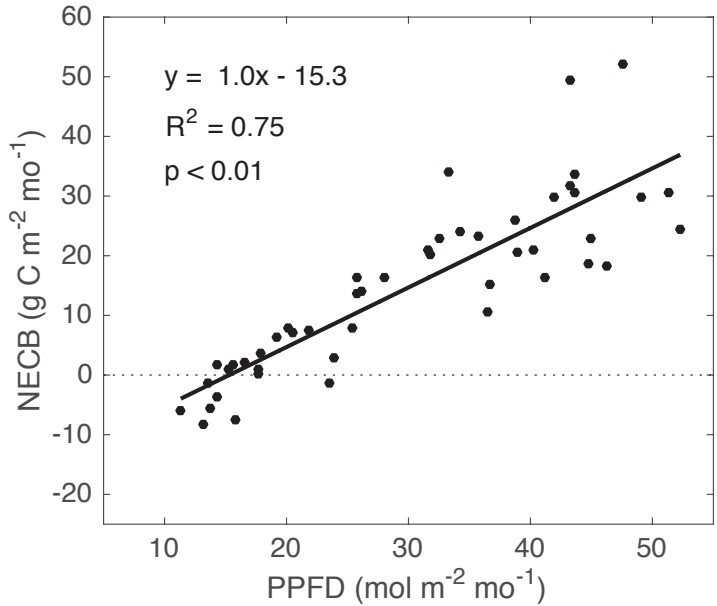

**Figure 6.** Monthly net ecosystem carbon balance (NECB) at Kopuatai bog as a function of monthly total PPFD from January 2012
5   to December 2015.



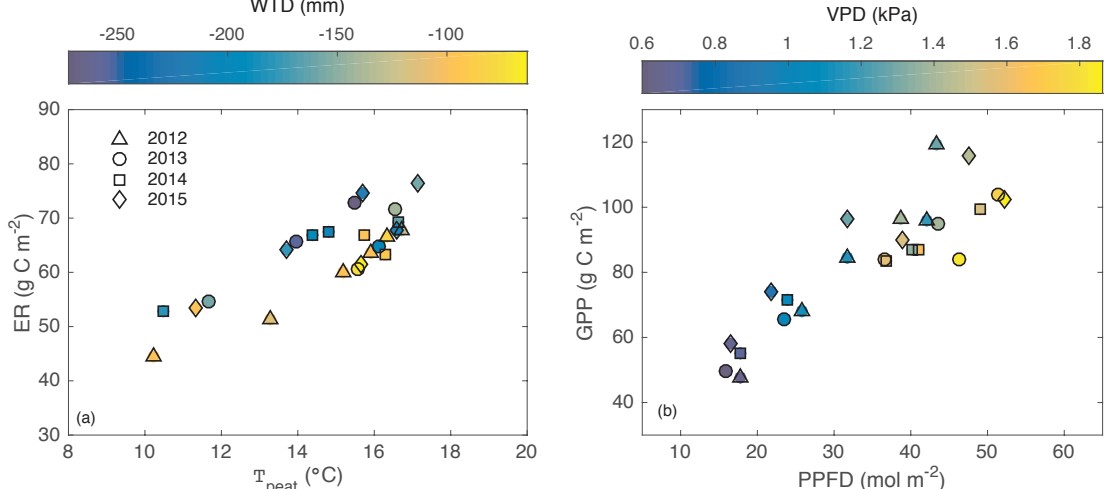

**Figure 7.** Summertime (Dec. – Feb.) and autumn (Mar. - May) monthly means of (a) ecosystem respiration (ER) versus peat temperature ($T_{peat}$) with symbol fill-colour according to water table depth (WTD), and (b) GPP versus integrated photosynthetic photon flux density (PPFD) with fill-colour according to daily maximum vapor pressure deficit (VPD) over the four measurement years.