# Peer review of "Southern hemisphere bog persists as a strong carbon sink during droughts"

_Biogeosciences, 2017_

## Referee Comment (RC1) · Anonymous Referee #1 · 3 May 2017

Review of Goodrich et al. BGD

General Comments

Goodrich et al. present results from an EC flux tower installed in a Empodisma robustum peatland over four years. Two of years have drought conditions for the summer period. The bog exhibited little sensitivity to the drought conditions and remained a C sink over the four years. Generally the paper is well written with a logical presentation. The work is very straight-forward as is the interpretation. I liked the paper and think it helps to fill in a whole in the literature for peatlands for regions outside the boreal. I found little fault with the study and anticipate it can be published with only minor revisions.

My main revision is to suggest a slight extension to the interpretation. I found the

response of 2012 vs. 2015 quite interesting. While we have a nice record showing the climatic conditions of 2012 - 2015, we don't have any data showing pre-2012 conditions. Obviously that is pretty normal, however I wonder about using the nearby weather station from NIWA to help interpretation for 2012. Since there is likely 4 years of overlapping meteorological data for the study bog and the NIWA station, what about using that to give a look at how the previous years conditions could have impacted 2012? Yes, this would need to be couched in cautious language and it would need to be recognized that the two sites are not going to have identical conditions but I think it could add to the interpretation of the early part of the bog record. I think the added uncertainty of this extrapolation from the NIWA site is reasonable and could be presented in a manner that acknowledges the added uncertainty.

Specific Comments

Abstract does not have GWP defined.

p 2 l 3 - the Bousquet study is not peatlands specific but rather wetlands as a whole.

p 3 l 23 - Spaghnum and other mosses...sentence, please consider rewording, bit confusing as written.

p. 5 l 2 - While I don't expect a full discussion on neural networks, some better lead to up to 'four fuzzy datasets representing season' etc. would help your readers. ANN and similar techniques are becoming more common but it is helpful to give readers a bit more background on the technique.

Sec 2.4 while I understand the DOC export paper is in preparation, it would be good to have more numbers here. E.g. p 5 l 30 'strong relationship', how strong? The DOC work presented needs to be able to stand on its own in this paper, especially as the other paper is not submitted nor published.

p. 8 l 31 - I didn't follow this argument about shifting away from E. robustum, what is the basis of this argument? Is it meant that the actual vegetation distribution would shift

due to ER demands?

p 11 l 16 - How deep is the peat?

Fig 7 - So each symbol represents one month right? That wasn't clear from the caption.

———————————————————

---

## Referee Comment (RC2) · Anonymous Referee #2 · 30 May 2017

The manuscript presented by Goodich et al. is an important study on the carbon budget of a Southern hemisphere peatland ecosystem. The authors quantified fluxes of CO2, CH4 and DOC over a period of four consecutive years, which makes it a valuable dataset to study effect of climate conditions on the peatland's carbon dynamics. In particular, the authors discuss changes in carbon fluxes during a strong drought year. By estimating global warming potential of the study area, they authors underline the bog's importance as a net carbon sink.

In the presented study, adequate methods are used to address the peatland's carbon dynamic. Eddy covariance fluxes of CO2 and CH4 were measured with state-of-the art instrumentation and the authors performed a detailed quality control assessment. In addition, the manuscript is well structured and written. Scientific points are presented in a conclusive manner and the paper is nice to read.

[Figure]

**BGD**

As the authors underline, there are little studies yet on the carbon dynamics of Southern hemisphere peatlands, which makes this study an important contribution. I suggest the manuscript to be published in Biogeosciences after addressing the two following issues:

1. As the authors state, the ecosystem respiration (ER) is positively correlated with the water table depth (WTD), although there might be a limit to the increase of ER with WTD. While the correlation between ER and WTD is not strong ($R^2 = 0.2$), it would be interesting to know what model function was used and if the authors considered using a different model? $CO_2$ transport through the peat column is not necessarily linear, deeper layers are probably more compacted, contributing less to the carbon exchange. The below surface $CO_2$ exchange is also governed by the vertical $CO_2$ profile in the column. This might suggest that ER does not increase linearly with increased gas volume (pore space).

2. A major focus of the manuscript is the discussion of GPP and ER for the different years. The authors mention a possible over-estimation of GPP and ER with the applied ANN method. As it is an essential part of the manuscript, can the authors give an estimate of this over-estimation or give an approximate error of the flux partitioning in relation to the measured $CO_2$ flux?

Other comments:

P. 3, L. 30: What was the height of the flux measurements (4.5 m or lower)? Do you have estimates of the corresponding footprint extent?

P. 4, L. 11: Göckede et al. (2006) use the scheme of Foken and Wichura (1996) in the revised version presented by Foken et al. (2004) for the flux quality assessment. As this scheme uses flags 1-9, which are the flags 1-5 you are referring to?

P. 4, L. 12: Here it would be interesting to state what the magnitude of the retrieved storage flux correction is.

[Figure]

P. 8, L. 3: PPDFD/VPD can be also interpreted as ratio. I suggest to provide an alternative here.

References:

Foken, T. and Wichura, B.: Tools for quality assessment of surface-based flux measurements, Agric. For. Meteorol., 78(1-2), 83–105, doi:10.1016/0168-1923(95)02248-1, 1996.

Foken, T., Göockede, M., Mauder, M., Mahrt, L., Amiro, B. and Munger, W.: Post-Field Data Quality Control, in Handbook of Micrometeorology, vol. 29, edited by X. Lee, W. Massman, and B. Law, pp. 181–208, Springer Netherlands., 2004.

Göckede, M., Markkanen, T., Hasager, C. B. and Foken, T.: Update of a footprint-based approach for the characterisation of complex measurement sites, Boundary-Layer Meteorol., 118(3), 635–655, doi:DOI 10.1007/s10546-005-6435-3, 2006.

---

## Author Comment (AC1) · 29 Jun 2017

**Southern hemisphere bog persists as a strong carbon sink during droughts [bg-2017-97]**

Goodrich, J.P., Campbell, D.I., Schipper, L.A.

Thank you for the opportunity to respond to the referees' comments on our manuscript. We were pleased that both referees considered this a valuable addition to the literature, and that they both envisaged only minor revisions. We are grateful for the attention that both referees gave.

Our proposed revisions are highlighted in blue.

**Referee #1:**

Referee #1's main suggestion involves using pre-2012 weather station data to aid in the interpretation of the early part of the bog flux data set. We have considered several potential avenues to address this suggestion, including looking into data from the nearby NIWA climate station, but ultimately feel that such an endeavor would constitute a much more major revision than the referee seemed to intend. The referee may be suggesting that we investigate the possibility of lagged effects of meteorological conditions on the flux response. However, this was not directly explored in the current manuscript, therefore would require substantially widening the scope of the study. The exceptional aspect of 2012 was the larger GPP during January. While lagged effects on ecosystem respiration and methane flux could conceivably play a role in the flux dynamics here (and we addressed some aspects of this with the hysteretic patterns evident in methane fluxes at Kopuatai in Goodrich et al., 2015), lagged effects on GPP are more difficult to attribute (e.g. Wiegand et al., 2004; Sherry et al., 2008). In the present manuscript, we show that most of the variability (83%) in GPP can be accounted for by a linear relationship to current PPFD and VPD (Table 2). Therefore, any additional lagged effect would be relatively subtle. Exploration of such effects may have merit, especially as the flux record at Kopuatai grows, but we feel this would change the nature of the present manuscript and require much more time than a minor revision would entail. We also note that Figure 1 includes water table depth and meteorological terms for December 2011, giving some additional context for the meteorological situation before the flux record begins. We will change the Figure 1 caption to clarify this.

Refs:

Goodrich, J.P., Campbell, D.A., Roulet, N.T., Clearwater, M.J., Schipper, L.A (2015) Overriding control of methane flux variability by water table dynamics in a Southern Hemisphere, raised bog. Journal of Geophysical Research, Biogeosciences, 120, 819-831.

Sherry, R.A., Weng, E., Arnone III, J.A., Johnson, D.W., Schimel, D.S., Verburg, P.S., Wallace, L.L, Luo, Y. (2008) Lagged effects of experimental warming and doubled precipitation on annual and seasonal aboveground biomass production in a tallgrass prairie. Global Change Biology, 14, 2923-2936.

Wiegand, T., Snyman, H.A., Kellner, K., Paruelo, J.M. (2004) Do grasslands have a memory? Modelling phytomass production of semiarid South African Grassland. Ecosystems, 7, 243-258.

***Referee #1's specific comments:***

P1 l25 (abstract): We propose expanding GWP to 'global warming potential' in the text.

P2 l3: We propose amending this sentence to the more accurate portrayal of recent studies on the global methane budget and contribution from tropical and Southern Hemisphere wetlands:

Tropical and Southern Hemisphere peatlands are particularly under-represented in the literature, despite contributing 10% of the global peatland area (Yu et al., 2010), and southern wetlands as a whole likely contribute >50% of the global wetland $CH_4$ budget (Bousquet et al., 2011; Bridgham et al., 2013).

P3 l23: We propose rewording this sentence to:

*Sphagnum* and other moss coverage is sparse throughout the peatland, occurring only where the dominant vegetation is relatively bare and light penetrates to the surface, and therefore the primary peat forming material is *E. robustum* roots (Agnew et al., 1993).

P5 l 2: We propose adding a sentence to follow the first mention of 'fuzzy datasets', which gives a description and direct citation for more info on their purpose:

These 'fuzzy datasets' are transformations of the decomposed time variables (year, season, month), which provide a way to avoid arbitrary accumulation of time information in the neural network (Papale and Valentini, 2003).

Sec. 2.4: We agree with this sentiment. In the text for this section, we propose adding the $R^2$ statistic (0.915) from the fit shown in the supplemental figure for this relationship. We felt that including Supplemental Figure 1 was critical given that the DOC-focused paper has yet to be published separately.

P 8 l31: We propose changing this sentence to clarify our intended meaning:

Our results suggest that lowered water tables increase ER at Kopuatai but there may be a limit to this increase. Further significant drops in water tables during severe drought may only be possible if the vegetation structure were to undergo a long-term shift away from *E. robustum*, with its conservative evaporation regime (Campbell and Williamson, 1997), to vegetation with higher water use (Thompson et al., 1999).

P 11 l16: We propose adding the peat depth to this sentence and simplifying to:

This is not a surprising conclusion considering the large annual NEP even during drought years and the very deep peat deposits (as deep as 14 m in places) that have accumulated during Kopuatai bog's ~10,000-year development (Newnham et al., 1995),…

We also propose adding a sentence to the end of Section 2.1 to note peat depth in the initial site description:

Peat depths at Kopuatai reach 14m, with an average peat accumulation rate of 0.9 mm yr$^{-1}$ throughout the Holocene (Newnham et al., 1995).

Fig 7 - Correct. We have re-arranged the caption to bring 'monthly means' to the front:

Monthly means of summertime (Dec. - Feb.) and Autumn (Mar. - May) (a) ecosystem respiration ….

**Referee #2:**

Again, we very much appreciate Referee 2's feedback and have attempted to address each comment/suggestion below:

1. The results shown in Table 2 are all from linear regression models, e.g. $y = m_1x_1 + b$ for a single regressor, and $y = m_1x_1 + m_2x_2 + b$ for dual regressors. The Referee is certainly correct in that the true relationship between ER and WTD will be complex and dependent on peat profile characteristics, however we used linear regression statistics as parsimonious diagnostics to compare basic relationships between the fluxes and what we expect to be their strongest controls. We propose clarifying this in the Table 2 caption describing the regression statistics:

**Table 2. Regression statistics for comparison of simple linear single- and dual-driver models explaining summer and autumn monthly ER and GPP over the four measurement years. Root mean square error (RMSE) and Akaike's information criterion (AIC) are given as measures of model error and relative quality, where a lower AIC value is favorable.**

2. We agree this is important to highlight, which is largely why we wanted to reference new published results. We do not have the direct surface respiration measurements or isofluxes necessary to estimate the potential discrepancies with the analytical approach we have used, and this is still a very active area of research among the flux community. However, we have added estimates of the potential bias when using neural-network or Reichstein-type partitioning methods in the proposed revision:

Oikawa et al. (2017) showed that results from flux partitioning based on neural networks behaved similarly to those based on the Reichstein et al. (2005) approach in an alfalfa field. However, both these approaches may overestimate GPP and ER (e.g. 10-13%, Oikawa et al., 2017) because they rely on extrapolating measured nighttime ER to daytime, whereas some studies have demonstrated lower plant respiration during daytime (Wohlfahrt et al., 2005; Wehr et al., 2016). The extent of this effect across all ecosystem types is unknown (Oikawa et al., 2017, Wehr et al., 2016), so interpretations based on partitioned ER and GPP should be stated cautiously. For this study, we applied the standard partitioning approach whereby nighttime GPP was assumed to be zero and daytime GPP was estimated by subtracting modelled daytime ER from gap-filled $F_{CO2}$.

Wehr, R., Munger, J.W., McManus, J.B., 2016. Seasonality of temperate forest photosynthesis and daytime respiration. Nature 534, 680–683.

Wohlfahrt, G., Bahn, M., Haslwanter, A., Newesely, C., Cernusca, A., 2005. Estimation of daytime ecosystem respiration to determine gross primary production of a mountain meadow. Agric. For. Meteorol. 130, 13–25.

***Other comments:***

P3 l30: We propose amending this section to specify that the instruments are mounted at 4.25 m above the peatland surface on the tower. Along these lines we feel it will also add value to provide an estimate of the footprint spatial extent since we suggest adequate fetch but give no quantification of this. We propose to add the following to this section:

Based on the analytical flux footprint model of Kormann and Meixner (2001), the average distance, centered on the EC tower, within which 80% of fluxes originated was 330 m.

Kormann, R., Meixner, F.X., 2001. An analytical footprint model for non-neutral stratification. Boundary-Layer Meteorology 99, 207–224.

P4 l11: The referee is correct and our description was incomplete. The EddyPro software preforms two main quality control checks including a steady state test and a test for well-developed turbulence (Foken et al., 2004), and provides standard choices for Mauder and Foken (2004) 0-1-2 flags, the Foken and Wichura (1996) and Gockede et al. (2006) 1-9 flags, or a simplified set of flags based on the same tests with fewer threshold categories to produce flags 1-5. We chose the intermediate detail as a way to ensure conservative quality control while avoiding unnecessary filtering of potentially useful data. We propose clarifying this in the text:

We utilized the composite EddyPro quality control flagging system (flags 1-5, with 1 being best quality) based on tests for steady state and well-developed turbulence conditions (Foken et al., 2004; Mauder and Foken, 2004; Foken and Wichura, 1996; and Göckede et al., 2006).

P4 l12: Maximum calculated storage fluxes were on the order of $0.25$ µmol m$^{-2}$ s$^{-1}$ during summer mornings, which we propose to explicitly state in the text here.

P8 l3: The referee is absolutely right! This awkward presentation will be changed to 'PPFD or VPD'.